# Effects of Daily Consumption of an Aqueous Dispersion of Free-Phytosterols Nanoparticles on Individuals with Metabolic Syndrome: A Randomised, Double-Blind, Placebo-Controlled Clinical Trial

**DOI:** 10.3390/nu12082392

**Published:** 2020-08-10

**Authors:** Yasna K. Palmeiro-Silva, Raúl I. Aravena, Lisette Ossio, Javiera Parro Fluxa

**Affiliations:** 1School of Nursing, Universidad de los Andes, Las Condes 7550000, Chile; lisette.ossio@gmail.com (L.O.); jfparro@uandes.cl (J.P.F.); 2Department of Chemical Engineering, Imperial College London, London SW7 2BX, UK; ria18@ic.ac.uk

**Keywords:** phytosterols, metabolic syndrome, waist circumference, triglycerides, lipoprotein, constipation

## Abstract

Metabolic syndrome (MS) affects up to 40% of the population and is associated with heart failure, stroke and diabetes. Phytosterols (PS) could help to manage one or more MS criteria. The purpose of this study was to evaluate the therapeutic effect of daily supplementation of an aqueous dispersion of 2 g of free-phytosterols nanoparticles in individuals with MS over six months of intervention, compared with placebo. This double-blind study included 202 participants with MS randomly assigned into phytosterol (*n* = 102) and placebo (*n* = 100) groups. Participants were assessed at baseline, 4, 12 and 24 weeks. General health questions, anthropometric measurements and blood parameters were analysed. At week 24, the proportion of participants with high triglycerides (≥150 mg/dL) in the phytosterol group was 15.65% lower than in the placebo group (*p*-value = 0.023). Similarly, half of the participants in the phytosterol group decreased their waist circumference up to 4 cm compared with 0 cm in the placebo group (*p*-value = 0.0001). We reported no adverse effects (diarrhoea or vitamin D reduction); nonetheless, almost 70% of participants in the phytosterol group self-reported an improvement in bowel habits. Daily intake of free-PS nanoparticles improved some MS criteria; therefore, it might be a promising adjuvant therapy for individuals with MS (NCT02969720).

## 1. Introduction

Metabolic syndrome (MS) is a cluster of interrelated physiologic and metabolic alterations that reaches a prevalence between 5% and 40% worldwide [1,2,3,4,5]. Despite different clinical definitions of MS, alterations in the lipidic profile, hyperglycaemia, obesity and high blood pressure are common characteristics [6,7,8,9,10,11,12]. Overweight, obesity and insulin resistance have been associated with MS as the main causes, which are strongly associated with diet and physical activity [13,14]. Complementary, MS and its components might increase short and long-term cardiovascular risks [15,16], partly because of cumulative macro- and microvascular changes as well as cellular dysfunction mediated by chemical and hormonal alterations [16,17]. These pathophysiological impairments increase the risk of heart failure, stroke and diabetes [18,19], representing a substantial burden of disease [20,21,22].

MS can be prevented or partially reversed by lifestyle changes and/or pharmacologic interventions usually focused on each MS criterion. The latter situation frequently leads to polypharmacy [23] and multiple adverse drug events, resulting in negative health outcomes and medication non-adherence [24]. In this regard, nutraceuticals have been proposed as natural alternatives to overcome these drawbacks, in which phytosterols (PS) or plant sterols are considered an adjuvant therapy for individuals with physiologic and metabolic alterations [25,26].

PS and phytostanols (PSn) are cholesterol-like molecules naturally found in vegetables and fruits. PS has been proposed to manage serum cholesterol since the 1950s when Pollak et al. reported that PS intake could reduce dietary cholesterol resorption [27]. Ever since then, several studies have reported that daily PS consumption may reduce serum levels of total cholesterol (TChol) and low-density lipoprotein cholesterol (LDL-c) [28,29,30]. Additionally, PS/PSn intake could improve other physiological and metabolic parameters. Plat et al. found that individuals who consumed two grams of PSn for 8-weeks reduced non-high-density lipoprotein cholesterol, TChol and triglycerides (TG) levels [31]. Sialvera et al. observed reductions in TChol, LDL-c, TG and small dense LDL amongst individuals with MS who consumed four grams of PS after two months of the intervention [32]. Conversely, another study demonstrated that intaking two grams of PS for three months did not improve the lipidic profile in individuals with MS [33]. Other studies have stated that PS could improve fasting glucose [34] and blood pressure [35].

Different factors could explain these diverse findings, for example, the food matrix in which PS/PSn is delivered, the daily dose of PS/PSn, free or esterified form, and population characteristics [36,37,38,39]. Particularly, free-PS nanoparticles (<1 µm) have shown hypotriglyceridaemic effects in comparison to PS esters in hypercholesterolaemic individuals [40]. However, to the best of our knowledge, no study has evaluated the effect of an aqueous dispersion of free-phytosterols nanoparticles (f-PSnano) in individuals with MS.

The primary purpose of this study was to evaluate the therapeutic effect of daily supplementation of an aqueous dispersion of 2 g of f-PSnano in individuals with MS over six months of intervention, compared with placebo. Secondary and exploratory analyses were conducted to expand our knowledge of the effects of PS consumption on other metabolic changes in individuals with MS.

## 2. Materials and Methods

### 2.1. Design

This study was a parallel, randomised, double-blind, placebo-controlled trial of f-PSnano in individuals with MS. This trial was conducted in three sites, considered as primary care centres, from January to November 2018, in Chile. Besides, this study adhered to the Declaration of Helsinki and Good Clinical Practice guidelines, was approved by the East Metropolitan Health Service Ethics Committee in Chile and registered at ClinicalTrials.gov (NCT02969720).

### 2.2. Participants

To be eligible, participants had to meet the following criteria: (i) age between 18 and 65 years; (ii) availability to attend morning appointments; (iii) have been diagnosed with metabolic syndrome or met diagnosis criteria at the beginning of the study. The individual is diagnosed with MS if she/he meets, at least, three out of five of the MS criteria (Table 1) proposed by Alberti et al. [13]. On the other hand, participants were not eligible if they: (i) had alcohol-related problems; (ii) had familial sitosterolaemia or hypercholesterolaemia; (iii) were consuming phytosterols at the moment of this study; (iv) were pregnant or breastfeeding; (v) had a medical history of myocardial infarction, stroke, decompensated diabetes or hypertension; (vi) not able to make decisions for themselves; (vii) people consuming weight management pills. During this study, there was no restriction on medication intake or dietary patterns.

The number of participants required was calculated considering two independent groups and dichotomous endpoint of success/failure, where success was considered starting with an abnormal criterion and finishing with a normal criterion (Table 1). Statistical parameters were a 1:1 enrolment ratio, alpha of 5%, power of 90% and 10% of the difference between groups. This calculation resulted in 190 participants, but we considered 15% of dropouts; thus, the final sample size was 110 participants per group.

### 2.3. Procedures and Assessments

This study included a screening period, one baseline visit (V1) and 24-weeks of the intervention phase, which included three visits (Figure 1).

Participants were screened from medical health records at the primary centres and selected if they fulfilled eligibility criteria for this study. If yes, they were contacted by the principal investigator (PI) and invited to the first meeting. If participants accepted, V1 was arranged and consisted of two phases. First, the PI explained the study in-depth and answered all participants’ questions. Once participants agreed to participate and provided written informed consent, they were randomly allocated to the intervention or placebo group. During the second part of V1, a nurse assessed participants according to the protocol and delivered one white box along with general instructions on how to consume the supplement at lunch (directly or diluted in a glass of water). Besides, a daily notebook was provided, in which participants could note any “extra” or “strange” symptom, missed doses or changes in diet or regular medication.

Assessments for all visits included: questions related to general health status; product tolerance; vital signs; anthropometric measures; blood tests; the analysis of participant’s notebook. In particular, vital signs included blood pressure (BP) and heart rate (HR), which were obtained using Omron 7120 monitors. Anthropometric measurements included height, weight and waist circumference (WC). Blood analysis included lipidic profile (TChol, LDL-c, very low-density lipoprotein cholesterol (VLDL-c), HDL-c, TG and lipid particles), biochemical profile (glycated haemoglobin (HbA1c), insulin, fasting glycaemia and homeostatic model assessment of insulin resistance (HOMA-IR) index) and vitamin D. Blood samples were collected after 12-h of fasting, in tubes containing clot activator and gel serum separator, sodium fluoride/potassium oxalate and ethylenediaminetetraacetic acid (EDTA), using a Vacutainer system^®^ or syringe when necessary. After collection, samples were placed in a blood sample container and taken to the ‘Instituto Radiologico Providencia’ in Chile.Fasting glycaemia and insulin were analysed using the glucose hexokinase (Glu-HK) and chemiluminescence methods, respectively; glycated haemoglobin was obtained through capillary electrophoresis method; lipid profile was obtained based on the cholesterol oxidase phenol 4-aminoantipyrine peroxidase (CHOD-PAP) method, and vitamin D was obtained using chemiluminescence method.

Bowel habit was evaluated based on self-perception and participant reports. Phone contact was maintained throughout the entire study according to the needs of participants or the research team.

At visit 2 (V2-week 4), visit 3 (V3-week 12) and visit 4 (V4-week 24), participants were assessed according to the protocol, and at V4 treatment, the allocation was revealed to each participant. Appendix A shows which assessment was performed at each visit.

### 2.4. Randomisation and Masking

Once enrolled, participants were allocated to the intervention group (phytosterol group) or placebo group in a 1:1 ratio according to a simple random sequence generated by STATA 13.0 software. The PI delivered coded white plain boxes containing placebo or f-PSnano sachets to the clinical research nurses. Participants and nurses were blinded.

### 2.5. Intervention

Participants allocated in the intervention group consumed an aqueous dispersion of 2 g of f-PSnano per day over six months period. Each sachet of dispersion (8 mL) contained f-PSnano from pine, surfactants (<0.5%) and water. Participants in the control group received an aqueous dispersion (10 mL) that consisted of titanium dioxide, xanthan gum, carrageenan, surfactants (<0.5%), potassium sorbate, citric acid and water. The difference of 2 mL between the product and placebo did not change the size of each sachet and was done to achieve the same white colour when phytosterols/placebo were consumed directly or diluted in a glass of water.

The aqueous dispersion of f-PSnano and placebo were manufactured and provided by Nutrartis S.A. Both were delivered in identical coded white plain cube boxes with 35 plain sachets, each one to ensure double-masking throughout the study.

### 2.6. Outcomes

The primary outcome was the proportion of participants classified with ‘abnormal’/’normal’ MS criteria at V4 between phytosterol and placebo groups. As individuals were not asked to suspend medication, including for MS, the proportion of participants ‘free’ of MS at the end of the study was not possible to evaluate. However, we considered the proportion of MS parameters that improved at V4 for both groups, which means going from ‘abnormal’ to ‘normal’. ‘Abnormal’ was understood as waist circumference ≥90 cm (men) or ≥80 cm (women); triglycerides ≥150 mg/dL; high-density lipoprotein cholesterol <40 mg/dL (men) or <50 mg/dL (women); systolic blood pressure ≥130 mmHg or diastolic blood pressure ≥85 mmHg; fasting glycaemia ≥100 mg/dL. Nonetheless, this binary classification could mask important differences in metabolic risk between individuals; therefore, we evaluated the metabolic syndrome severity Z score (MetS-Z), based on waist circumference, proposed by Gurka et al. [41,42], in order to obtain a clearer degree of the overall metabolic syndrome severity. It is worth noting that this score was created based on the USA population and needs to be validated to the Chilean population.

Major secondary outcomes were values of total cholesterol (TChol), LDL-c, very-low-density lipoproteins (VLDL-c), high-density lipoproteins (HDL-c), triglycerides (TG), fasting glycaemia (FG), glycated haemoglobin (HbA1c), fasting insulin (FI), HOMA-IR index, weight, body mass index (BMI), waist circumference (WC), systolic and diastolic blood pressure (SBP and DBP, respectively) at V2, V3 and V4, as well as the difference between each visit and baseline, between groups. Exploratory outcomes included the particle number of lipoproteins for a subgroup of participants at each visit. Safety outcomes included adverse and serious events. Before starting the study, diarrhoea was considered as a potential adverse effect because PS consumption could modify bowel habits due to changes in cholesterol absorption. Besides, Vitamin D was measured because of a possible alteration in lipophilic vitamin absorption [43]; therefore, they were included as routine assessments. Any event that required medical or surgical intervention due to the intervention was considered a serious adverse event.

### 2.7. Statistical Analyses

Each MS criterion was considered as a categorical variable (‘normal’/’abnormal’) according to cut-off points presented in Table 1, and the metabolic syndrome severity score was calculated according to Gurka et al. [41,42]. Complementary, MS criteria and secondary outcomes were analysed as continuous variables in order to observe more precise changes and the magnitude of change from the baseline because some participants were taking medications, a situation that could not be reversed; thus, baseline values matter. Subgroups analyses were conducted to explore sex differences (women/men). Exploratory outcomes were analysed descriptively.

Data normality was assessed using summary measures, histograms and the Shapiro–Wilk test. For normally distributed data, mean and standard deviation (SD) are presented, and parametric tests were used (Student’s T-test); otherwise, median and 25th and 75th percentiles (p25–p75) were considered, and non-parametric tests were applied (Mann–Whitney U test). When comparing categorical data, Chi-square test or exact Fisher’s test were used, and the test of proportions was applied accordingly.

In the case of non-heterogeneous groups, we adjusted effects for sex, age and basal conditions using generalised linear regression models.

For all tests, the level of significance was set at less than 5%, and all statistical analyses were performed using STATA 13.0.

## 3. Results

### 3.1. Sample Characteristics

From medical records, we screened 850 individuals, of whom 546 were not eligible for not fulfilling MS criteria. A total of 219 participants were enrolled and randomly assigned to either phytosterol (*n* = 109) or placebo (*n* = 110) arm. Throughout the study, 17 participants dropped out mostly because they did not have enough time to attend morning assessments. The cumulative frequency of dropout was the following: 12 participants dropped out at V2 (six from placebo and six from phytosterol), 14 participants at V3 (seven from placebo and seven from phytosterol) and 17 participants at V4 (ten from placebo and seven from phytosterol). Despite these dropouts, 202 participants completed this study with 100% of data. Figure 2 shows the flow chart of the study.

Baseline participants’ characteristics were well balanced between groups, not having significant differences regarding sociodemographic, health-related and physiologic variables at baseline (Table 2).

### 3.2. Primary Outcome

At the beginning of this study, participants met between three and five MS criteria (Table 2), and the distribution was similar between groups (table *p* = 0.379). At V4, some participants changed the status of these criteria just based on blood tests, WC and BP measurements, considering that no regular medication was suspended or changed. Table 3 shows the distribution of MS criteria met by participants at V4 and the difference in this number between V4 and V1.

Considering the analysis of each MS criterion, the proportion of participants with high or ‘abnormal’ TG levels (≥150 mg/dL) between phytosterol and placebo groups were similar at baseline, reaching 2.06% of the difference (*p*-value = 0.7691); however, this proportion was significantly different at V4, reaching 15.65% of the difference between groups (*p*-value = 0.023). This difference was explained by a decrease in participants with high TG levels in the PS group and an increase in the placebo group. For all other criteria, the proportion of participants with ‘abnormal’ levels at V4 was not statistically different between groups (Figure 3). For detailed information, see Appendix A.

Table 4 presents each MS criterion considering continuous data. At visit 4, WC was significantly different between groups, where half of the participants in the phytosterol versus placebo group presented WC up to 94 and 99 cm, respectively (*p* = 0.0022). Other small differences were observed in TG and HDL-c, but they were not statistically significant.

In terms of the difference between V4 and V1 for each parameter, WC and TG were significantly different between groups. Half of the participants in the phytosterol group decreased WC up to 4 cm in comparison with 0 cm in the placebo group, which means that participants in the phytosterol group outperformed placebo, decreasing WC by 5.05%. Similarly, half of the participants in the phytosterol group decreased TG levels up to 16.5 mg/dL. This finding means that the phytosterol group outperformed placebo group, showing a 15.17% reduction in TG levels.

The metabolic syndrome severity was evaluated through the MetS-Z based on waist circumference. At V1, this score was similar for both groups (*p* = 0.5947), but at V4, this score decreased by 0.199 (25th percentile: −0.559; 75th percentile: 0.106) for the phytosterol group and increased by 0.009 (25th percentile: −0.301; 75th percentile: 0.25) for the placebo group, this difference being statistically significant (*p* = 0.0024).

At V4 and based on generalised linear models, we observed that participants in the phytosterol group had, on average, 3.76 cm less WC than participants in the placebo group, adjusting for age, sex and WC baseline values (*p*-value = 0.001 and 95% confidence interval from −5.1 cm to −2.4 cm). Similarly, participants in the phytosterol group presented, on average, 27.49 mg/dL less of TG levels than the placebo group, after adjusting for age, sex and TG baseline levels (*p*-value = 0.004 and 95% confidence interval from −46.26 mg/dL to −8.73 mg/dL).

### 3.3. Secondary Outcomes

Secondary outcomes included analysis of lipidic profile (TChol, LDL-c, VLDL-c, HDL-c, TG), FG, FI, HOMA-IR index, weight, BMI, WC, SBP and DPB at V1, V2, V3 and V4. Baseline values are shown in Table 2, and detailed values for each variable at each visit are shown in Appendix A.

We observed that participants in the phytosterol group showed consistently better results for TChol, VLDL-c, TG, HOMA-IR index and WC than the placebo group. The relative difference of TChol levels and HOMA-IR index at each visit (V2, V3, V4) in comparison with baseline levels was consistently lower for the phytosterol group than the placebo group, but this difference was only statistically significant between V2 and V1 (*p* = 0.012 and *p* = 0.0211, respectively) (Appendix A). Analysing the average reduction, participants taking phytosterols reduced TChol levels by 3.3%, while participants in the placebo group increased TChol levels by 1.9% at the end of the first month of intervention (Figure 4a).

Similarly, the phytosterol group presented consistent and systematic reductions of VLDL-c than the placebo group at each visit compared with baseline. The median relative reduction in VLDL-c amongst participants in the phytosterol group was 5.6%, 7.7% and 10.5% when comparing V2, V3 and V4 with baseline values, respectively. In contrast, these values for the placebo group were close to 0% at all times (Figure 4b).

In the same way, TG levels followed a similar trend to that of VLDL-c. Participants in the phytosterol group had significant reductions in TG compared with the placebo group at all times. In Figure 4c, we observe that half of the participants in the phytosterol group reduced TG levels by 7.9% between V2 and V1; 9.7% between V3 and V1; 12.6% between V4 and V1, while the placebo group showed a change of almost 0%. This difference was statistically significant at all times (Appendix A).

Finally, waist circumference showed statistically significant results when comparing relative differences between V3 and V1 and V4 and V1 (Figure 4d). At V3, half of the participants in the PS group had reduced WC by 2.09%, while this reduction was 0% in the placebo group (*p*-value = 0.0007). At V4, this difference was even larger. Half of the participants in the placebo group presented a change of 0% from the baseline, while this reduction in the phytosterol group reached 3.74% (*p*-value = 0.0001).

### 3.4. Safety Outcomes

Vitamin D levels were similar between groups at baseline and V4. For results at baseline, see Table 2. The median for vitamin D at V4 was 18.1 ng/mL for the phytosterol group (25th percentile: 14.1 ng/mL; 75th percentile: 23.8 ng/mL) and 17.6 ng/mL for the placebo group (25th percentile: 13.85 ng/mL; 75th percentile: 23.4 ng/mL), showing a not statistically significant difference (*p*-value = 0.5313).

When consulting for bowel habits, no participant reported diarrhoea episodes during this study. However, most of the participants in the phytosterol group self-reported an improvement in their bowel habits at V4, a situation that was not observed in the placebo group (Table 5). According to participants’ description, either in the placebo or phytosterol group, most of them self-reported a previous state of “constipation” due to hard or dry stools; however, at visit 4, almost 70% of participants in the phytosterol group described an improvement in this condition, relieving constipation but not presenting diarrhoea (*p*-value = 0.001).

### 3.5. Exploratory Outcomes

In a random subsample of participants, we explored the effects of phytosterols on the particle number for VLDL (VLDL-p), LDL (LDL-p) and HDL (HDL-p) and their different subclasses (small, medium and large) at each visit. These parameters were analysed for 59 participants from the phytosterol group and 57 participants from the placebo group. Socioeconomic, physiological variables and baseline values of the particle number were similar in both groups; therefore, we concluded that they were comparable (Appendix A).

We observed that at V4, participants in the phytosterol group tended to have lower LDL-p and VLDL-p than the placebo group (Figure 5). For detailed values, see Appendix A.

## 4. Discussion

In this study, we evaluated the therapeutic effect of daily supplementation of an aqueous dispersion of 2 g of f-PSnano in individuals with MS over six months of intervention, compared with placebo. We observed that the phytosterol group reduced significantly TG levels and WC after six months of intervention. These results are of importance because high TG levels and WC have been widely reported as critical risk factors for cardiovascular diseases [44,45,46,47,48,49,50,51] and metabolic disorders [52,53]. Additionally, based on the MetS-Z, the phytosterol group decreased its metabolic syndrome severity in comparison with the placebo group, which means that individuals who consume phytosterols could decrease their metabolic syndrome severity. However, this score needs to be standardized and validated to the Chilean population.

Hypotriglyceridaemic effects of PS have not been consistent in previous studies. Naumann et al. [54], Demonty et al. [55] and Rideout et al. [56] established that TG levels reduction was highly dependent on TG levels at baseline, being the reduction greater in individuals with higher TG baseline levels. In particular, we observed a difference of 15.17% in TG levels between groups at V4, which was consistent with the results reported by Plat et al. [31] and Sialvera et al. [32] in individuals with MS.

Although the hypotriglyceridaemic effect of PS is not well understood, some mechanisms have been proposed to explain it [31,32,57]. Our exploratory analyses showed that participants who consumed f-PSnano tended to have lower VLDL-p compared with the placebo group, which agreed with Plat et al., who reported that individuals with MS presented a significant reduction in large and small VLDL-p after PSn esters consumption compared with placebo [57]. It is well-known that TG is mainly carried by VLDL-p, whereby TG levels reduction could be mediated by a reduction in the hepatic synthesis of VLDL-p; however, more lipoprotein synthesis studies are needed to confirm this hypothesis.

To the best of our knowledge, this is the first study that showed constipation relief and reduction of WC amongst participants who consume PS. A decrease in intestinal cholesterol absorption due to cholesterol displacement from the mixed micelle [58,59] could explain these results. In this regard, we hypothesised that hypotriglyceridaemic effects, constipation relief and WC reduction after consumption of f-PSnano could be driven by a reduction in fatty acid absorption.

High fatty acid absorption could lead to an increment in WC by an increment in visceral adiposity (VA). Several studies have proposed that higher WC is positively correlated with a higher amount of VA in individuals [60,61,62]. A higher amount of VA tends to increase the lipolytic activity of this tissue [63], producing a high triglyceride turnover in VA tissue [64]. As VA strongly influences lipid metabolism [65,66,67,68], VA increments could promote fatty acids delivery to the liver through the portal vein [69] because of a higher TG turnover. Consequently, an increment of VA could lead to a higher TG transportation to and concentration in the liver, triggering an increase in VLDL-c production [65,70,71]. Therefore, a reduction in fatty acid absorption could lead to a reduction in the VA tissue and a reduction in the efflux of fatty acids from the VA to the liver, decreasing liver TG concentration. Then, the lower availability of TG in the liver could lead to a reduction in VLDL-c production and exportation to the vascular system, reducing the serum TG level. Complementary, the reduction of intestinal fatty acid absorption would improve constipation by changing the faeces composition.

Regarding the well-known hypocholesterolaemic effects of PS consumption, we observed that TChol levels tended to be lower in the phytosterol group, observing significant changes between groups only at V1. These findings could be explained by three main reasons. First, most of the participants in this study were normo-cholesterolaemic; thus, only slight changes in TChol and LDL-c would be expected [38,54]. Second, long-term clinical trials have reported minor TChol reductions in normo-cholesterolaemic and mild-hypercholesterolaemic individuals who consumed PS in comparison with short-term trials [72,73,74]. This observation could explain why Plat et al. [31] and Sialvera et al. [32] reported a significant reduction in TChol and LDL-c after 8-weeks of intervention, while Hernández-Mijares et al. reported non-significant effects over these parameters after 12-weeks of the intervention [33]. Finally, PS reduces TChol and LDL-c by decreasing cholesterol absorption, but individuals with MS present low cholesterol absorption and high endogenous cholesterol synthesis [75]; therefore, the PS lowering absorption effect could be compensated with the endogenous synthesis of cholesterol in individuals with MS in long-term studies. Indeed, increased endogenous cholesterol synthesis has been observed in hypercholesterolaemic individuals under PS intervention [76,77].

Interestingly, vitamin D levels decreased in both groups throughout the study, showing no significant difference between groups at V4. We explained this phenomenon as a seasonality effect because we started recruiting in summer, and V3 and V4 were performed during fall-winter seasons.

Individuals with MS exhibit higher VLDL-p and LDL-p in comparison with healthy individuals [78]; therefore, we explored the effects of f-PSnano consumption on lipoprotein particle number. Our results showed that VLDL-p and LDL-p concentration tended to be lower in individuals who consumed f-PSnano compared with the placebo group. As it was mentioned before, reduction in VLDL-p could be related to reductions in TG and WC because of reduced fatty acids absorption. Although lower LDL-p in the phytosterol group in comparison with placebo group could be explained by the hypocholesterolaemic and hypotriglyceridaemic effects of PS, we were not able to explain why LDL-p increased in both groups along the study. We hypothesised that it could be related to changes in dietary habits or physical activity throughout the year. It is worth noting that small LDL-p was lower in the phytosterol group at V2, which coincided with a reduction of TCHol, VLDL-c and TG levels.

This study has several strengths. It is the first study that evaluated f-PSnano in participants with MS. It also included a larger sample size compared to previous studies and was done according to a real routine clinical practice. To the best of our knowledge, this is the first study that reported a reduction in WC due to f-PSnano consumption, an anthropometric parameter that is highly associated with cardiovascular risk. Finally, this study showed an improvement in bowel habit; therefore, f-PSnano could be an interesting solution for many individuals who have physiologic and metabolic alterations, as well as constipation, without adverse effects on fat-soluble vitamins. Nonetheless, this study has potential limitations. It considered only three primary centres, so, representativeness and generalisation of results could be limited. Besides, the sample was mainly integrated by women, which could be a bias in terms of sex representation. This drawback was addressed performing regression analyses. Finally, this study covered six months of intervention, and long-term effects of f-PSnano consumption were not evaluated.

## 5. Conclusions

In conclusion, this study showed that an aqueous dispersion of f-PSnano could be an interesting adjuvant therapy in individuals with MS. Besides, more studies are needed to evaluate long-term effects and other potential effects on physiologic and metabolic alterations.

## Figures and Tables

**Figure 1 nutrients-12-02392-f001:**
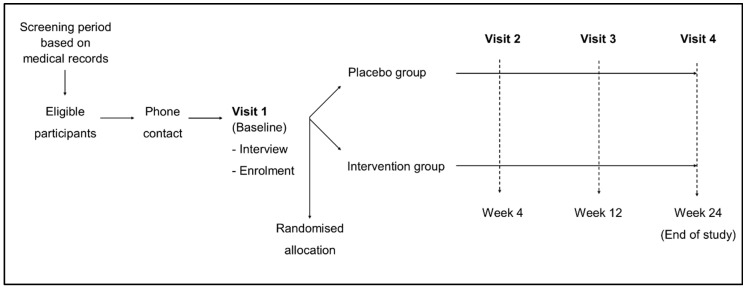
General diagram of the study.

**Figure 2 nutrients-12-02392-f002:**
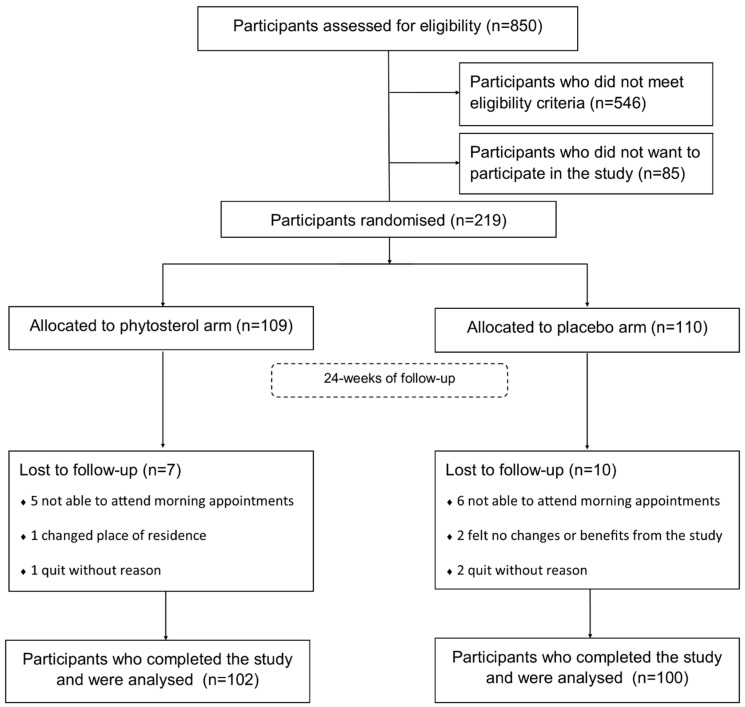
Flow chart of the study.

**Figure 3 nutrients-12-02392-f003:**
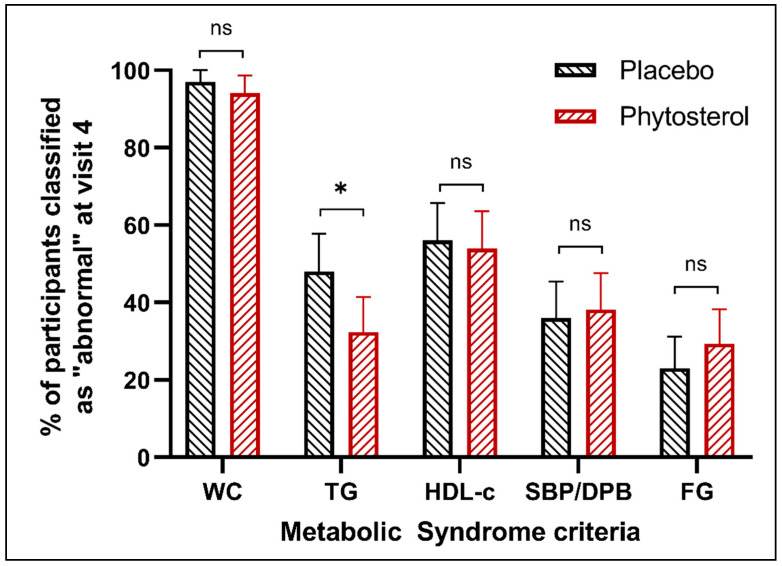
Distribution of participants (%) with ‘abnormal’ levels of waist circumference, triglycerides, high-density lipoprotein cholesterol (HDL-c), systolic/diastolic blood pressure and fasting glycaemia at V4 by the group. ns = non-significant; * statistically significant (*p* < 0.05).

**Figure 4 nutrients-12-02392-f004:**
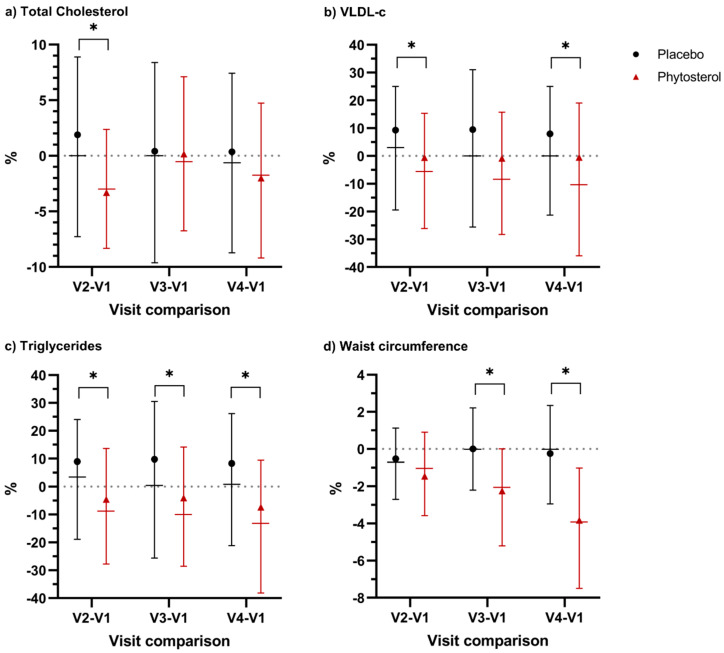
Relative difference (%) between V2 and V1, V3 and V1 and V4 and V1 for (**a**) total cholesterol; (**b**) very-low-density lipoprotein cholesterol (VLDL-c); (**c**) triglycerides; (**d**) waist circumference by the group. * Statistically significant (*p* < 0.05). Black lines represent the placebo group, and red lines the phytosterol group. Horizontal lines represent the interquartile range and the median. Black points (•) represent the mean for the placebo group, and red triangles (▲) represent the mean for the phytosterol group.

**Figure 5 nutrients-12-02392-f005:**
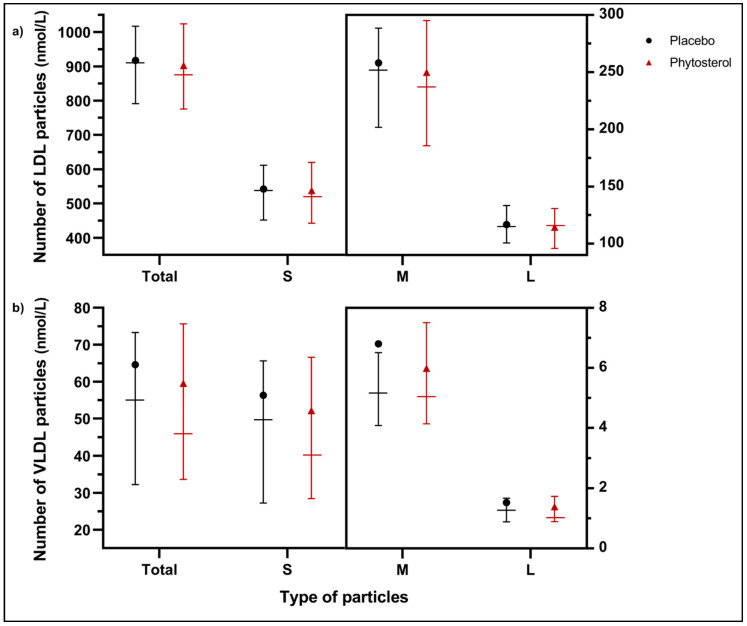
Number of LDL-p (**a**) and VLDL-p (**b**) by subclass (small, medium and large) at V4 by the group. Black lines represent the placebo group, and red lines the phytosterol group. Horizontal lines represent the interquartile range and the median. Black points (•) represent the mean for the placebo group, and red triangles (▲) represent the mean for the phytosterol group. M and L particles are measured with the right y-axis.

**Table 1 nutrients-12-02392-t001:** Metabolic syndrome (MS) criteria agreed in 2009.

Criterion	Categorical Cut-Off Point
Waist circumference	Depends on ethnic and country-specific definitions ^1^
Triglycerides	≥150 mg/dL or specific treatment (fibrates or nicotinic acid)
High-density lipoprotein cholesterol (HDL-c)	<40 mg/dL in males; <50 mg/dL in females, or specific treatment (fibrates or nicotinic acid)
Blood pressure (BP)	Systolic BP ≥ 130 or Diastolic BP ≥ 85 mmHg, or treatment of previously diagnosed hypertension
Fasting glycaemia	≥100 mg/dL, or previously diagnosed with type 2 diabetes

^1^ South American population is considered for this study: ≥90 cm (males) and ≥80 cm (females). Source: [13].

**Table 2 nutrients-12-02392-t002:** Baseline participants’ characteristics by the group.

KERRYPNX	Phytosterol (*n* = 102)	Placebo (*n* = 100)	*p*-Value
Sociodemographic variables			
Age in months, median (p25–p75)	520 (407–636)	527.5 (414–632.5)	0.9568
Sex			
Men, *n* (%)	28	26	
Women, *n* (%)	74	74	0.816
Socioeconomic status (self-reported)			
High, *n* (%)	3 (2.94)	2 (2)	
Middle, *n* (%)	93 (91.18)	90 (90)	
Low, *n* (%)	6 (5.88)	8 (8)	0.733
General health-related information			
Underlying health condition			
Yes, *n* (%)	86 (84.31)	79 (79)	
No, *n* (%)	16 (15.69)	21 (21)	0.329
Concomitant medication			
Yes, *n* (%)	63 (61.76)	57 (57)	
No, *n* (%)	39 (38.24)	43 (43)	0.491
Smoking			
Yes, *n* (%)	55 (53.92)	44 (44)	
No, *n* (%)	47 (46.08)	56 (56)	0.158
Alcohol consumption			
Yes, *n* (%)	67 (65.69)	72 (72)	
No, *n* (%)	35 (34.31)	28 (28)	0.333
Number of meals per day, median (p25–p75)	3 (3–3)	3 (3–3)	0.1777
Physical activity			
Yes, *n* (%)	44 (43.14)	46 (46)	
No, *n* (%)	58 (56.86)	54 (54)	0.682
Blood test parameters, median (p25–p75)			
Total cholesterol (mg/dL)	196 (169–224)	196 (168.5–225)	0.9789
LDL-c (mg/dL)	118.5 (99–144)	118.5 (100.5–147.5)	0.8069
VLDL-c (mg/dL)	26.5 (20–42)	28.5 (20–40.5)	0.9625
Insulin (µUl/mL)	15.6 (10.38–23.7)	13.36 (9.07–18.48)	0.1065
HOMA-IR index	3.02 (1.94–4.26)	3.62 (2.28–5.52)	0.0684
Glycated haemoglobin - HbA1c (%)	5.215 (4.98–5.77)	5.305 (4.96–5.69)	0.7361
Vitamin D (ng/mL)	24.6 (19.9–29)	24.25 (19.7–29.55)	0.9109
Anthropometric measurements and vital signs, median (p25–p75)			
Weight (kg)	78.95 (71.2–90.1)	80.6 (70.5–89.75)	0.8985
Height (m)	1.585 (1.55–1.67)	1.59 (1.545–1.675)	0.7413
Body mass index (kg/m^2^)	30.63 (27.46–34.4)	30.55 (28–34.46)	0.5298
Heart rate (beats per minute)	72 (67–80)	71 (65–76.5)	0.0455
Metabolic syndrome variables, median (p25–p75)			
Waist circumference (cm)	99.5 (93–105)	100.5 (92–108.5)	0.4795
Triglycerides (mg/dL)	138 (105–221)	143.5 (100.5–202.5)	0.8303
HDL-c (mg/dL)	43.5 (37–52)	43 (37.5–51)	0.9875
Systolic blood pressure (mm Hg)	122 (113–133)	119.5 (110.5–132)	0.2176
Diastolic blood pressure (mm Hg)	83 (78–90)	82 (76–89)	0.6188
Fasting glycaemia (mg/dL)	93.5 (88–99)	91 (85.5–98)	0.0910
Metabolic syndrome criteria, *n* (%)			
3	71 (69.61)	76 (76)	0.379
4	22 (21.57)	20 (20)	
5	9 (8.82)	4 (4)	
MetS-Z, median (p25–p75)	0.463 (0.06–0.99)	0.489 (−0.02–0.93)	0.5947

(p25–p75): 25th percentile and 75th percentile; LDL-c: low-density lipoprotein cholesterol; VLDL-c: very-low-density lipoprotein cholesterol; HDL-c: high-density lipoprotein cholesterol; MetS-Z: metabolic syndrome severity Z score.

**Table 3 nutrients-12-02392-t003:** Distribution of MS criteria met by participants at V4 and the difference between V4 and V1.

*N*° of Criteria	Visit 4			*N*° of Criteria	Difference Between V4 and V1		
	Phytosterol (*n* = 102)	Placebo (*n* = 100)	*p*-Value		Phytosterol (*n* = 102)	Placebo (*n* = 100)	*p*-Value
0, *n* (%)	2 (1.96)	1 (1)	0.563	−3, *n* (%)	3 (2.94)	1 (1)	0.648
1, *n* (%)	16 (15.69)	16 (16)		−2, *n* (%)	24 (23.53)	19 (19)	
2, *n* (%)	39 (38.24)	30 (30)		−1, *n* (%)	45 (44.12)	40 (40)	
3, *n* (%)	29 (28.43)	35 (35)		0, *n* (%)	23 (22.55)	31 (31)	
4, *n* (%)	9 (8.82)	14 (14)		1, *n* (%)	6 (5.88)	8 (8)	
5, *n* (%)	7 (6.86)	4 (4)		2, *n* (%)	1 (0.98)	1 (1)	

**Table 4 nutrients-12-02392-t004:** MS criteria (continuous data) at visit 4 and the absolute difference between V4 and V1 by the group. All data are presented as median and 25th–75th percentiles.

	Visit 4			Difference Between V4–V1		
	Phytosterol (*n* = 102)	Placebo (*n* = 100)	*p*-Value	Phytosterol (*n* = 102)	Placebo (*n* = 100)	*p*-Value
WC (cm)						
All	94 (89–102)	99 (93–109.5)	0.0022	−4 (−7–−1)	0 (−3–2.5)	0.0001
Men	99.5 (94.5–105)	103 (94–109)	0.5732	−3 (−5–0)	0 (−3–3)	0.0875
Women	92 (86–99)	99 (93–110)	0.0007	−5 (−8–−1)	0 (−3–2)	0.0001
TG (mg/dL)						
All	123 (87–175)	145 (99.5–191.5)	0.1137	−16.5 (−57–15)	1.5 (−40–32)	0.0245
Men	150 (123–243.5)	188 (122–245)	0.4566	−21.5 (−87–3.5)	−37 (−54–13)	0.9724
Women	112.5 (82–152)	134.5 (86–168)	0.1176	−14 (−57–17)	3.5 (−32–32)	0.0074
HDL-c (mg/dL)						
All	46 (40–55)	45 (39–53.5)	0.6839	1 (−1–5)	1 (−2–5)	0.6563
Men	39.5 (35.5–46)	37.5 (31–45)	0.3721	0 (3.5–−2)	0 (−3–2)	0.5037
Women	48 (42–58)	47 (41–55)	0.6071	2 (−1–6)	2 (−2–5)	0.8102
SBP (mm Hg)						
All	119 (112–129)	119 (112.4–128)	0.9242	−2 (−12–5)	−1 (−10.5–7.5)	0.2753
Men	125 (118–131)	127.5 (117–136)	0.6093	0 (−12.5–7)	2.5 (−9–8)	0.7095
Women	116.5 (112–126)	118 (108–126)	0.7735	−2.5 (−12–4)	−1 (−11–7)	0.2939
DBP (mm Hg)						
All	79.5 (73–87)	80 (72.5–86)	0.6820	−2.5 (−9–3)	−3 (3.5–−10)	0.9099
Men	85 (78.5–91)	84 (75–91)	0.6398	−2 (2.5–−9)	−5 (−15–2)	0.3630
Women	78 (72–85)	79 (70–85)	0.9480	−3 (−9–3)	−2.5 (−10–5)	0.4508
FG (mg/dL)						
All	95 (90–102)	92 (87–98)	0.1712	2 (−3–7)	1.5 (−3–6)	0.9769
Men	96 (90–99.5)	94 (92–105)	0.7947	2 (−2–12)	3.5 (0–7)	0.9378
Women	94.5 (86–103)	91.5 (85–97)	0.1179	1 (−4–6)	1 (−3–6)	0.9480

WC: waist circumference; TG: triglycerides; HDL-c: high-density lipoprotein cholesterol; SBP: systolic blood pressure; DBP: diastolic blood pressure; FG: fasting glycaemia.

**Table 5 nutrients-12-02392-t005:** Participants’ perception regarding bowel habits at V4 by the group.

	Phytosterol (*n* = 102)	Placebo (*n* = 100)
Bowel habit improved from V1, *n* (%)	69 (67.65)	4 (4)
Bowel habit did not change from V1, *n* (%)	31 (30.39)	90 (90)
Bowel habit worsened from V1, *n* (%)	2 (1.96)	6 (6)

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
