# Peer review of "Effects of Daily Consumption of an Aqueous Dispersion of Free-Phytosterols Nanoparticles on Individuals with Metabolic Syndrome: A Randomised, Double-Blind, Placebo-Controlled Clinical Trial"

_nutrients, 2020, doi:10.3390/nu12082392_

Round 1

Reviewer 1 Report

The authors of this double blind clinical trial evaluated the therapeutic effect of daily supplementation of an aqueous dispersion of 2 grams of free-phytosterols nanoparticles in patients with Metabolic Syndrome over six months of intervention, compared with placebo. The statistical analysis of study can be improved if the authors carry out followings:

  1. Since all the patients had metabolic syndrome at the time of enrollment, and since the primary outcome of study is metabolic syndrome, and since metabolic syndrome is a reversible condition, it is important to report and compare how many patients were free of metabolic syndrome in both groups at the end of study.
  2. Since metabolic syndrome is a categorical variable, it would be more precise if the authors calculated metabolic syndrome severity score for each patient and compare this reliable index between the two groups and also within groups, before and after intervention.

Reviewer 2 Report

The aim of this study was to evaluate the therapeutic effect of daily supplementation of an aqueous dispersion of 2 grams of free-phytosterols nanoparticles in patients with MS over six months of  intervention, compared with placebo.

 In my opinion this manuscript is  interesting but  suffers from some flaws. I enumerate them in the order of appearance:

  1. The subject and purpose of this work (line 2 and 66) is related to patients with metabolic syndrome (MS). In the description of the study group MS factors were listed (Table 1 criteria from 2009), but it was not specified how many factors are necessary to diagnose MS? Did all the participants of the study meet at least 3 criteria of MS?
  2. Do research participants are patients (hospitalization at baseline point)? Maybe it's better to use the term: individuals or participants.
  3. In publication reference no. 40, the authors used water dispersible formulation of free plant sterols (WD-PS). The WD-PS, sub-micron dispersion of free sterols with the targeted composition and particle size was prepared by Nutrartis S.A. (Santiago, Chile), using the patent application WO 2010/095067 25. WD-PS was added to 100 g yogurt (2 grams WD_PS in 100 g yogurt).

In this publication (line 130) participants allocated in the intervention group consumed an aqueous dispersion of 2 grams of f-PSnano from  Nutrartis S.A. (Santiago, Chile).

In my opinion water dispersible formulation and aqueous dispersion is the same product form.

For that reason please explain the sentence (line 63): “However, to the best of our knowledge, no study has evaluated the effect of an aqueous dispersion of free-phytosterols nanoparticles (f-PSnano) in patients with MS”?

  1. Please present the instruction, how the participants consumed the liquid from the sachets with PS and placebo group.
  2. Subjects from intervention group and control group (placebo) should use the same-looking preparation. Why is there a difference in volume (8 ml and 10 ml) (line 131 and 133)?
  3. Please include blood sampling and clinical measurement’s procedures.
  4. Please include methodology of laboratory analyses.
  5. Please calculate HOMA-IR index.
  6. Line 229 – the results were adjusted for sex and age (the Phytosterol and Placebo group were well balanced for age and sex – Table 2, why the adjustment for age and sex  was calculated?).
  7. In my opinion it may be good to calculate the number of participant with diagnosed MS (≥3 factors of MS) in Phytosterol and Placebo group at visit 4 (as a Primary outcome related to MS).
  8. Figure 4 -please include the name of parameters (eg: total cholesterol) on the figure (the figure will be more readable).
  9. In my opinion, the title of the manuscript and the aim should not be focused on MS but should also include the assessment of individual lipid fractions. The authors present a lot of results for the lipid profile including number and size of lipoprotein particles. The primary purpose was only specified in the manuscript.  It would be worth defining the aim for secondary and exploratory outcomes.All outcomes  should also be finally summarized (finally conclusion).
  1. Table 2SM – please verify the calculation of difference of proportion between groups at V4 https://www.medcalc.org/calc/comparison_of_proportions.php
  2. Table 2SM n – it may be interesting to calculate the difference of proportion between Phytosterols V4 and baseline and Placebo V4 and baseline.
  3. Table 6SM and Table 7SM do not contain p values for differences? It would be worth to calculate it.
  4. The term “ the therapeutic effect ” (line 66 and 302) is inappropriate. Therapeutic effect refers to the response after a treatment. In this case it would be better to use the term: “lipid lowering effect” or “the effects on lipids or MS parameters”.
  5. It may be interesting to calculate the effect on TG concentration at V4 point separately in group with TG> and < 150 mg/dl at baseline.
  6. Whether this study has the status of a clinical trial? (line 321). If not please use the term “ study”
  7. Line 324“constipation relief has not been reported in patients consuming PS” from which study (reference) is this result?
  8. In this study you have found that nano-PS consumption was associated with the higher number of LDL-p, especially small LDL-p, compared with  healthy individuals (line 358). Studies show that people whose LDL particles are predominantly small and dense have a greater risk of coronary heart disease. Please discus this problem eg.  Nutr Metab Cardiovasc Dis. 2012 Oct;22(10):843-8. doi: 10.1016/j.numecd.2010.12.004.
  9. The discussion section is sometimes not very clear (it is not clear whether the results presented in the discussion refer to this study or other studies)

Round 2

Reviewer 1 Report

Thank You!

Author Response

Thank you. We appreciate very much your feedback and revision.

Reviewer 2 Report

Thank you very much for responding to all my comments.

  1. Please, complete the methodology for  laboratory tests (comment no. 7), e.g. name of the laboratory where the analyzes were performed, names of manufacturers of chemical reagents or biochemical analyzer, names of laboratory methods.
  2. I think that Figure 4 is duplicated line 325
  3. Table 3, column 5 (No of criteria: -3, -2...) line 236; what do the numbers mean?

Author Response

Thank you for your comments. According to your feedback:

  1. Please, complete the methodology for  laboratory tests (comment no. 7), e.g. name of the laboratory where the analyzes were performed, names of manufacturers of chemical reagents or biochemical analyzer, names of laboratory methods.
    • Answer: Thanks for your comment. We have included more details in relation to laboratory tests (Line: 127-130). However, the names of manufacturers of chemical reagents were not included because it would compromise third parties (enterprises) that are not directly related to this study. The name of the laboratory appears in the "acknowledgements" section.
  2. I think that Figure 4 is duplicated line 325
    • Answer: Figure 4 appears duplicated because the manuscript is using tacking changes, so if you see, the first N4 figure appears with a horizontal red line, which means that has been deleted. Then, the second N4 figure is the final figure which includes the name of each plot (as you suggested in your first revision).
  3. Table 3, column 5 (No of criteria: -3, -2...) line 236; what do the numbers mean?
    • Answer: As you have suggested before, we incorporated the number of MS parameters at V1 and V4. In table 3, we are presenting the distribution of MS criteria met by participants at V4 and the difference of these criteria between V4 and V1 (column 5 represent the difference between V4 and V1).